# Crop Classification and Representative Crop Rotation Identifying Using Statistical Features of Time-Series Sentinel-1 GRD Data

Xin Zhou ⓘ, Jinfei Wang *ⓘ, Yongjun He ⓘ and Bo Shan

Department of Geography and Environment, The University of Western Ontario, London, ON N6A 3K7, Canada
* Correspondence: jfwang@uwo.ca

**Abstract:** Compared with a monoculture planting mode, the practice of crop rotations improves fertilizer efficiency and increases crop yield. Large-scale crop rotation monitoring relies on the results of crop classification using remote sensing technology. However, the limited crop classification accuracy cannot satisfy the accurate identification of crop rotation patterns. In this paper, a crop classification and rotation mapping scheme combining the random forest (RF) algorithm and new statistical features extracted from time-series ground range direction (GRD) Sentinel-1 images. First, the synthetic aperture radar (SAR) time-series stacks are established, including VH, VV, and VH/VV channels. Then, new statistical features named the objected generalized gamma distribution (OGΓD) features are introduced to compare with other object-based features for each polarization. The results showed that the OGΓD $\sigma^{VH}$ achieved 96.66% of the overall accuracy (OA) and 95.34% of the Kappa, improving around 4% and 6% compared with the object-based backscatter in VH polarization, respectively. Finally, annual crop-type maps for five consecutive years (2017–2021) are generated using the OGΓD $\sigma^{VH}$ and the RF. By analyzing the five-year crop sequences, the soybean-corn (corn-soybean) is the most representative rotation in the study region, and the soybean-corn-soybean-corn-soybean (together with corn-soybean-corn-soybean-corn) has the highest count with 100 occurrences (25.20% of the total area). This study offers new insights into crop rotation monitoring, giving the basic data for government food planning decision-making.

**Keywords:** crop classification; crop rotation; objected generalized gamma distribution (OGΓD); time series; Sentinel-1; synthetic aperture radar (SAR)

## 1. Introduction

Compared with a monoculture planting mode, the practice of crop rotations can improve fertilizer efficiency and increase crop yield [1]. However, although crop rotations have been practiced for thousands of years, crop rotation patterns at a large scale are rarely identified due to the lack of fundamental data such as crop-type maps. The traditional ground survey can output high-accuracy crop-type distributions, but it consumes a large workforce and a lot of time. Remote sensing technology is an alternative way, providing long-term and large-scale crop-type information by crop classification, which can also achieve crop-changing monitoring and crop rotation mapping [2,3].

Some studies demonstrated the feasibility of using annual crop maps to make the crop rotation analysis. Panigraphy et al. [2] used a maximum likelihood algorithm and four-date multispectral data to map crop rotations in India. Even though the proposed method can acquire the rice plant pattern, the rotation information derived from two-year data was unreliable. Sahajapal et al. [4] proposed a Representative Crop Rotations Using Edit Distance (RECRUIT) algorithm to select representative crop rotations by combining and analyzing multi-year crop digital layers. However, the rotation mapping result using the cropland data layer (CDL) may have limited accuracy due to the uncertainty of the CDL products. Waldhoff et al. [3] used multi-temporal multispectral remote sensing data,

ancillary information, and expert knowledge of crop phenology to realize crop mapping from 2008 to 2015, which provided basic data for analyzing representative crop rotation patterns in Germany. Li et al. [5] equipped a crop rotation identification approach using MODIS and Landsat data by combining the random forest classifier and an improved flexible spatiotemporal data fusion (IFSDAF) model for multiple growing seasons of crops, achieving an overall accuracy of 90%. Recently, Liu et al. [6] proposed a hybrid neural network framework to identify crop rotation maps by synergizing time-series Sentinel-1 and Sentinel-2 images. However, the deep-learning method cannot give detailed explanations of the crop rotation process.

The weather conditions and cloud cover may hinder the performance of optical sensors to discover crop rotations. Synthetic aperture radar (SAR), an active microwave sensor without the limitations of illumination or weather conditions by its penetration capability, provides a more reliable approach for long-term monitoring of crop type changes and crop rotation [6]. Early studies focused on using single-frequency SAR data for crop classification [7,8]. Compared with the single-polarization mode, the multi-polarization SAR, including full-polarization (containing HH, HV, VH, and VV) and dual-polarization modes, achieved better classification performance than the single-mode SAR due to their different sensitivities to crop canopies and structures [9–11]. The ratios of polarization channels and scattering mechanisms derived from polarimetric target decomposition, such as surface scattering, double-bounce scattering, and volume scattering, were reported to help identify grain crops [12–14]. In addition, Frate et al. [15] reported that phase information helped with crop classification. Although many studies have explored crop classification methods using a single temporal SAR image, their accuracies were limited because phenological information on crop growth stages was ignored.

The multi-temporal or time-series SAR images have been successfully tested on crop classification in recent years. Backscatter variations in multi-temporal SAR images are composed of changes in crop canopy, crop structure, and soil physical parameters, which help users discriminate various crop types [16]. Skriver [17] compared different polarization combinations and frequencies via the Wishart classifier, and the Hoekman and Vissers classifier showed similar performance of crop classification when using C-band and L-band SAR images. Blaes et al. [18] combined multi-temporal SAR with Landsat TM images to achieve crop identification. The experimental results showed that the SAR data could improve the classification accuracy compared with using optical satellites alone [18]. Bargiel [19] introduced a description of phenological sequence patterns using dense Sentinel-1 images to achieve a more robust classification for different crop types and farming management conditions. However, the method requires priority information on phenological stages for each crop type, which is challenging when applied in a large-scale region.

Some studies combined machine-learning and deep-learning methods with polarization signatures of SAR images to achieve better classification results. The decision trees and multi-temporal SAR images were combined to develop new crop classification methods, showing an improvement in accuracy compared to single-date data [20,21]. Gao et al. [22] introduced a novel pairwise proximity function support vector machine (ppfSVM) classifier for time-series dual-polarization SAR data, that simultaneously considers the local shape of the curve and the temporal range of the crops. By contrast, deep-learning methods require more training data than traditional machine-learning approaches, which showed a higher accuracy when using both SAR and optical images [23,24]. Mestre-Quereda et al. [25] compared the radiometric and interferometric features of the dense Sentinel-1 time series and found that they both output good results, and their combination achieved the best results.

Another type of target description of SAR images is called statistical features, which does not rely on the scattering mechanism of individual pixels, improving the robustness of classification results. Whelen [26] introduced the coefficient of variation (CV) as a unitless measurement for cropland classification using SAR images. However, the CV can only discriminate between crop and non-crop regions, which cannot satisfy the requirements of precision agricultural monitoring. It is reported that [27] a new superpixel-based statistical

texture feature was proposed, called $G^0$ statistical feature, combined with polarimetric features to classify cropland in Canada. However, the $G^0$ statistical feature can only be extracted from fully polarimetric SAR images, limiting its application in other polarimetric modes such as dual-polarization or single-polarization. Therefore, to improve crop classification accuracy, it is necessary to develop effective statistical features for more practical data, such as Sentinel-1 GRD data, which are open access and free to download.

In this paper, a crop classification and rotation mapping scheme combining the random forest (RF) algorithm and new statistical features, named objected generalized gamma distribution (OGΓD) features, extracted from time-series ground range direction (GRD) Sentinel-1 images, is proposed. First, the SAR time-series stacks based on Sentinel-1 data are established, including VH, VV, and VH/VV polarizations. Then, the object-based features for those three channels can be calculated, including backscattering features and the OGΓDs. After comparing the classification performance of these features combined with the RF model, the one that achieves the highest accuracy is selected to output the annual crop map for each year. Finally, annual crop-type maps of five consecutive years (2017–2021) are synthesized as a crop sequence dataset for identifying representative crop rotation patterns.

## 2. Study Area and Data

### 2.1. Study Area

The study area is near London city in southwest Ontario, Canada (Figure 1), with three typical crop types: corn, soybean, and wheat. Other ground cover types include grass, woodland, and build-up. The crop types in each field changed every year due to crop rotation. Generally, corn and soybean are seeded in May and harvested in October. Winter wheat in this study site is planted in October of the previous year and harvested in July of the following year [23].

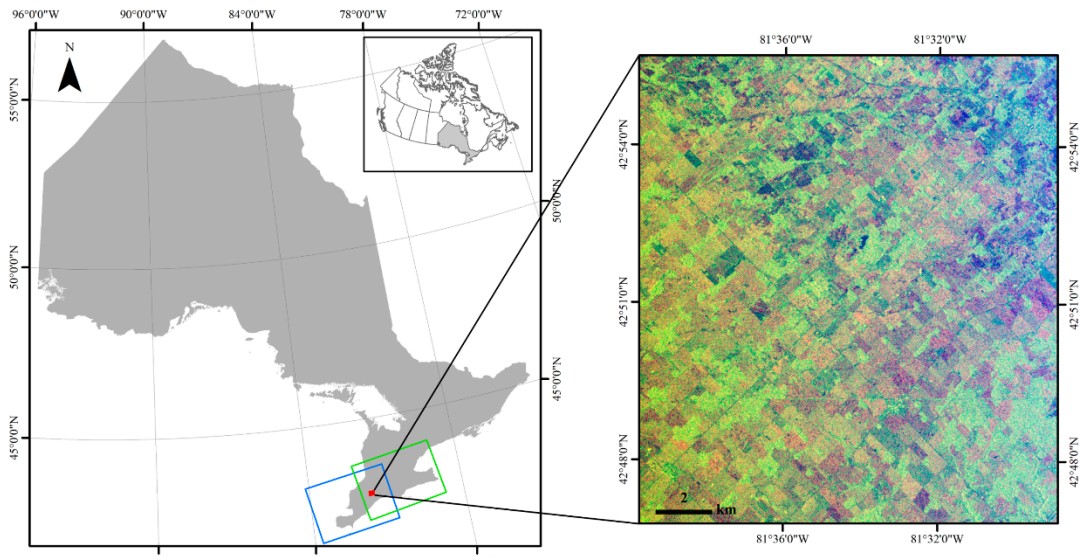

**Figure 1.** Study area location and Sentinel-1 data. Two Sentinel-1 orbits cover the study area. The green one is orbit number 77 with an incidence angle of 31.5°. The blue one is orbit number 150 with an incidence angle of 41.5°.

### 2.2. Data

The ground truth of crop types from 2017 to 2021 for model training and validation was collected based on ground survey and visual interpretation, as shown in Figure 2. First, the crop types and the locations of their fields were recorded and converted to vector layers in ArcMap software by drawing polygons using optical satellite images as the base

map. Then, for each year, 100 polygons were selected as the training dataset and the other 100 polygons were used as the validation dataset.

**Figure 2.** Ground truth data for crop classification and validation from 2017 to 2021.

The satellite data used in this study are Sentinel-1 ground range direction (GRD) SAR data. Time-series Sentinel-1 data with the dual-polarization mode (VH and VV) are provided by the European Space Agency (ESA), covering the whole crop growing season between 1 May and 30 November from 2017 to 2021. The data acquisition time of each scene is shown in Figure 3. Due to our study area being located in an overlap of two satellite orbits (orbit number 77 with an incidence angle of 31.5° and orbit number 150 with an incidence angle of 41.5°), the data access frequency is less than the standard revisit time of Sentinel-1. Some dates were missed due to the changes in the acquisition plan. The Sentinel-1 data were downloaded from Google Earth Engine (GEE), and had been pre-processed using thermal noise removal, radiometric calibration, and terrain correction, and converted to a raster grid with a spatial resolution of 10 m.

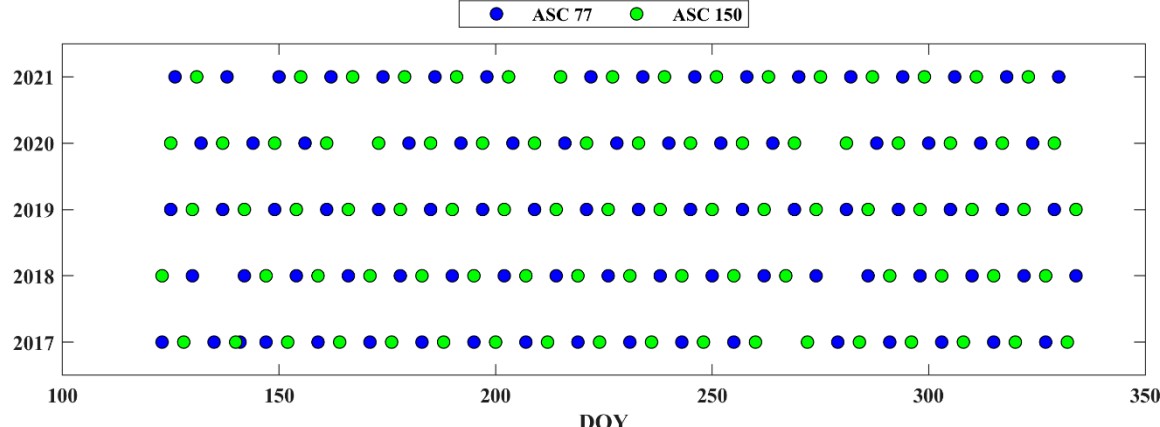

**Figure 3.** Sentinel-1 data acquisition date. The *x*-axis is the day of the year (DOY) and the *y*-axis indicates the years. ASC represents the ascending orbit.

## 3. Method

The workflow of crop classification and representative crop rotation identification using statistical features of time-series Sentinel-1 GRD data is shown in Figure 4, including data preparation, object-based feature calculation, crop classification, and crop rotation

analysis. In addition to VH and VV polarization, the ratio of them (VH/VV) which has been tested in previous agricultural applications, is also used in this research [28]. Object-based features, including object-based backscatter and OGΓDs, are estimated on the field scale. The RF classifier evaluates these features and their combinations to determine the best performance. Then, high-accuracy annual maps of crop types can be obtained based on the feature and the RF model. Finally, crop rotation sequences and representative crop rotation patterns are analyzed based on annual crop classification results.

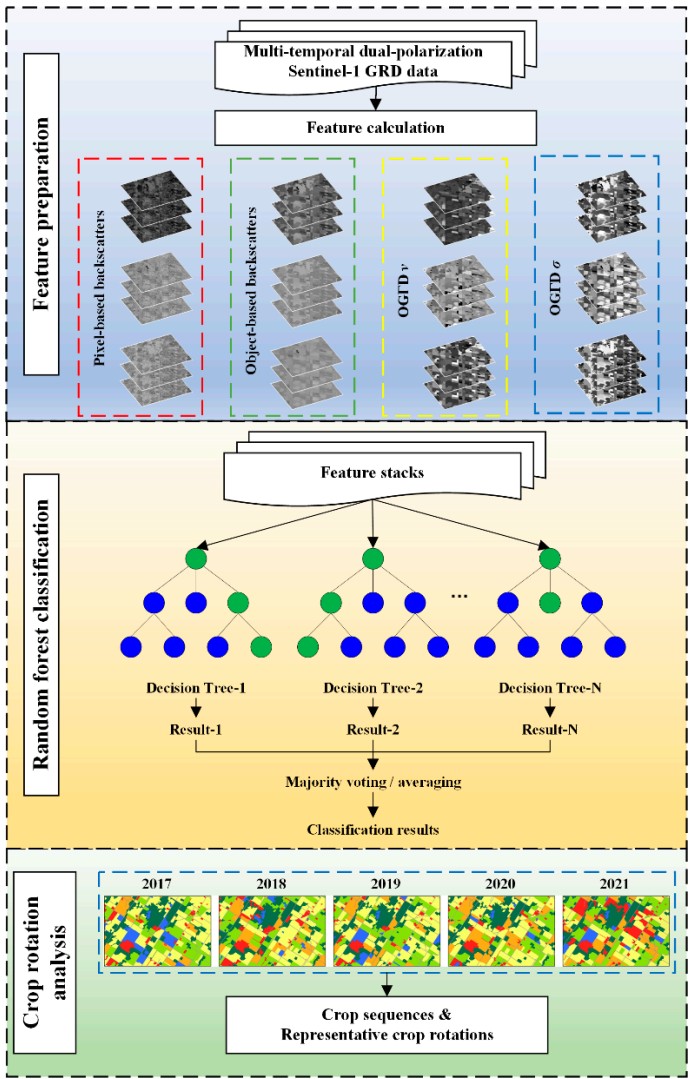

**Figure 4.** The workflow of the crop classification and representative crop rotation identification using statistical features of time-series Sentinel-1 GRD data.

### 3.1. Generalized Gamma Distribution (GΓD)

The inherent speckle noise usually hinders the image interpretation of SAR images. However, on the other hand, these speckles also provide information, as different land cover types exhibit differences in statistical properties. The empirical model is established by analyzing the characteristics of real SAR data and finding a mathematical probability distribution function (PDF) to describe them. Compared to physical models, although empirical models lack mechanism explanations, they have been successfully used for classification and target detection [29]. The generalized gamma distribution (GΓD) is

one of the most reliable empirical models that can be applied in both homogeneous and heterogeneous regions. The PDF of the GΓD was defined in [30] as

$$p(z) = \frac{|v|k^k}{\sigma\Gamma(k)}\left(\frac{z}{\sigma}\right)^{kv-1}\exp\left\{-k\left(\frac{z}{\sigma}\right)^v\right\}\sigma, |v|, k > 0, z \geq 0 \tag{1}$$

where $\Gamma(\cdot)$ is the gamma function, $\sigma$ is the scale parameter, $v$ is the power parameter, and $k$ is the shape parameter.

In order to estimate the three parameters, the method of log-cumulant (MoLC) is introduced, which is similar to the moment estimation method. Firstly, the log-cumulant equations of the GΓD model are established, making the overall log-cumulant of the model equal to the log-cumulant of samples. Then the parameters of the model can be obtained by solving the equations. For the samples $\mathbf{X} = \{X_1, X_2, \dots, X_N\}$, the $r$th-order estimators of log-cumulants were defined as

$$\begin{cases} \hat{c}_1 = \frac{1}{N}\sum\limits_{i=1}^{N}\ln X_i \\ \hat{c}_r = \frac{1}{N}\sum\limits_{i=1}^{N}(\ln X_i - \hat{c}_1)^r, \ r \geq 2 \end{cases} \tag{2}$$

where $N$ is the number of samples involving the parameter estimation. The first three orders of the log-cumulants can be used for estimating the parameters of the GΓD, written as

$$\frac{\Psi^3\left(1, \hat{k}\right)}{\Psi^2\left(2, \hat{k}\right)} = \frac{\hat{c}_2^3}{\hat{c}_3^2} \tag{3}$$

$$\hat{v} = \mathrm{sgn}(-\hat{c}_3)\sqrt{\Psi\left(1, \hat{k}\right)/\hat{c}_2} \tag{4}$$

$$\hat{\sigma} = \exp\left\{\hat{c}_1 - \left(\Psi\left(\hat{k}\right) - ln\hat{k}\right)/\hat{v}\right\} \tag{5}$$

where $\Psi(\cdot)$ and $\Psi(m, \cdot)$ are the digamma function and the mth-order polygamma function, respectively, and $\mathrm{sgn}(\cdot)$ is the sign function.

In Equation (3), the estimator $\hat{k}$ is determined by the $\hat{c}_2^3/\hat{c}_3^2$, while the $\hat{v}$ and $\hat{\sigma}$ are determined by the $\hat{k}$ according to Equations (4) and (5). In order to solve Equation (3), the dichotomy that is a simple numerical method needs to be applied. After that, the other two parameters $\hat{v}$ and $\hat{\sigma}$ can also be estimated. Since the $\Psi^3\left(1, \hat{k}\right)/\Psi^2\left(2, \hat{k}\right)$ is a continuous monotonically increasing function, a unique minimum of 0.25 is obtained when $\hat{k}$ approaches zero. Therefore, the ratio of second-order and third-order log-cumulant is required to satisfy $\hat{c}_2^3/\hat{c}_3^2 \geq 0.25$, otherwise the estimator $\hat{k}$ would fail to estimate using the numerical method. An approximation method for solving the problem when $\hat{c}_2^3/\hat{c}_3^2 < 0.25$, is written as the following [30].

$$\frac{\hat{k}^2}{\hat{k} + \frac{1}{2}} = \frac{\hat{c}_2^3}{\hat{c}_3^2} \tag{6}$$

Due to the two different equations for the estimation of the shape parameter ($\hat{k}$), it is not as reliable as the other two parameters. In addition, the $\hat{k}$ derived from Equation (3) has a large dynamic range. Thus, only the scale parameter ($\hat{\sigma}$) and power parameters ($\hat{v}$) will be used in this study for the subsequent crop classification.

### 3.2. Crop Classification Using the RF Classifier

One of the challenges for crop classification using SAR images is speckle noise, which is an inherent property of SAR sensors. Object-based features are introduced in this study to suppress the impact of speckle noise on classification performance. Usually, the

object-based feature requires image segmentation to generate segmentation objects, such as superpixel methods [27]. However, the performance of different segmentation methods may impact the subsequent crop classification results. For example, some misclassified pixels in the boundary area may be due to the inaccurate segmentation boundary rather than the classification process, which means that the errors in the image segmentation step are passed on to the final classification results. In order to avoid the influence of image segmentation on the classification, segmentation objects used in this study are generated through manual digitalization based on optical images such as Sentinel-2 and Planet images.

Once the segmentation objects are generated, the medium values of backscatter in each object are calculated and combined within the classification input sets named object-based backscatter. The object-based backscatter includes all date data and three polarization modes: VH, VV, and VH/VV channels. In addition, OGΓDs were also calculated based on these segmentation objects. Each segmentation object is viewed as an estimation window, in which the three parameters of the GΓD are estimated using all of the objects' pixels.

The classifier used in this study is a machine-learning method named random forest (RF), which is an ensemble of many individual decision trees. Compared with other machine-learning methods, such as maximum likelihood (ML) or support vector machine (SVM), the RF is insensitive to hyperparameters. Although the SVM and the ML are also acceptable, we only tested the RF model because the primary purpose is to explore the potential of statistical information for crop classification and rotation mapping, rather than to develop a new machine-learning model. In addition, the RF model not only has a good performance in identifying different crop types, but also it can output the importance of input features, which provides evidence for us to evaluate their contributions. The range of feature importance is from 0 to 1, and a larger value indicates a more significant feature. In addition to the OGΓDs mentioned previously, pixel-based and object-based backscatter will be used as the input of the crop classification for comparison purposes. Moreover, the combination of these three types of features will also be explored to discover the optimal feature set.

For the validation of crop classification results, four assessment metrics are introduced, including producer's accuracy (PA), user's accuracy (UA), overall accuracy (OA), and Kappa coefficient. PA is the probability that the certain land cover of an area on the ground is classified as such, and UA is how often the class on the map will be present on the ground. For evaluating the performance of single features and feature combinations, the UA is used for each crop type and the OA and Kappa are used for the overall assessment. For the accuracy of each year, all four metrics are used.

After obtaining the classification results for all five years, the crop rotation sequence can be obtained and the representative rotation pattern can be analyzed. The segmentation objects generated by the manual are used for crop rotation analysis on the field scale. Here, the crop sequence means a five-year array that indicates the whole changing process of crop types in each field, while crop rotation is a subset of the crop sequence that generally occurs in a two- or three-year cycle. Therefore, as long as the crop sequence is obtained, the typical crop rotation pattern can be analyzed.

## 4. Experiment and Results

### 4.1. Crop Classification Using a Single Type of Feature

Four types of features were tested in this part, including pixel-based backscatter, object-based backscatter, OGΓD $v$, and OGΓD $\sigma$. Since 2019 has the most complete data of the five years, 36 scenes of SAR images in 2019 were selected for the accuracy evaluation. Examples of the features calculated based on SAR images taken on 4 July 2019 are shown in Figure 5. For each feature type, three polarization channels were included: VH, VV, and VV/VV polarization. After the feature set preparation, all features are inputted into the RF classifier for model training and crop classification. The training and validation datasets are shown in Figure 2. Each year, there are 100 parcels for training and 100 other parcels

for validation. Therefore, the number of trees of the RF classifier is set at 100 to ensure the performance of the model.

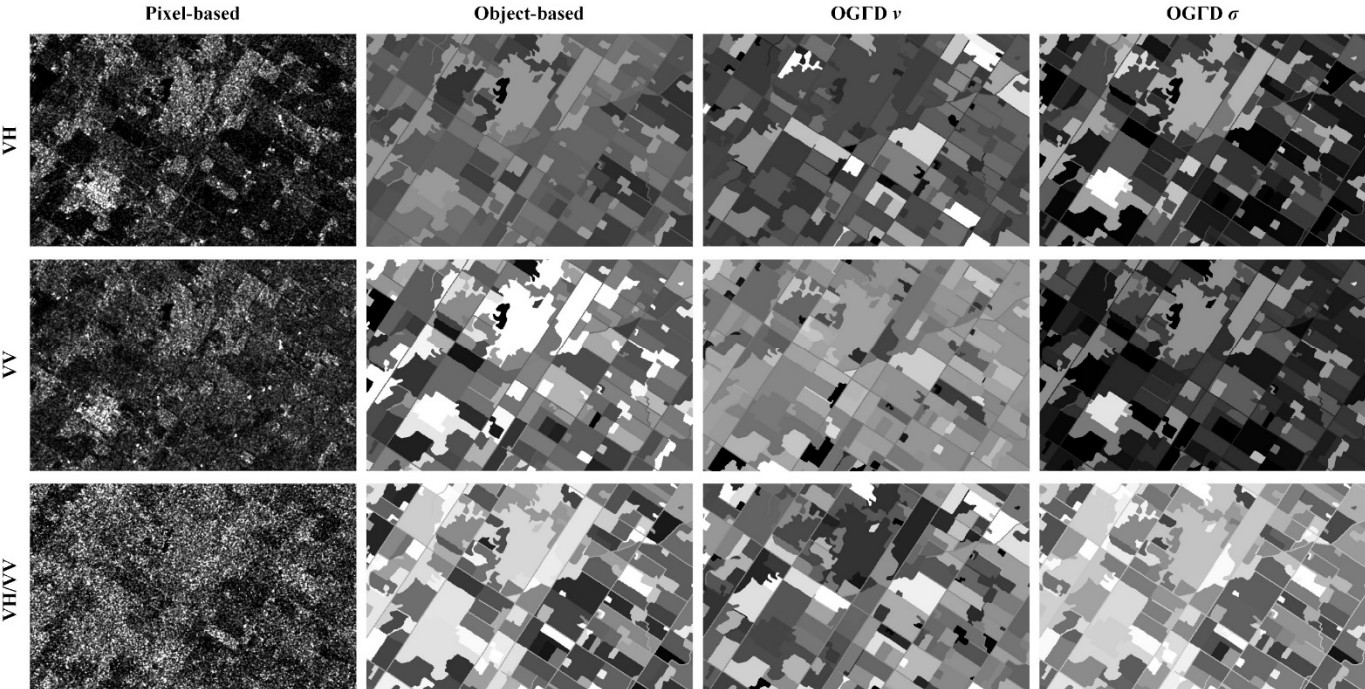

**Figure 5.** Examples of features calculated based on SAR images taken on 4 April 2019. Each column represents a different feature type and each row represents a polarization channel.

Figure 6 shows the crop classification results using the single type of feature, and Table 1 shows the assessment metric results. Obviously, the classification results of pixel-based backscatter have much more speckle noise than the other three types of object-based features. The quantitative evaluation results showed that pixel-based classification only achieved 85.21% of the OA and 79.19% of the Kappa in VH polarization. The classification errors of build-up regions hindered the precision of pixel-based backscatter. In contrast, the results of object-based backscatter performed better. The classification results of VH polarization achieved 92.17% of the OA and 89.16% of the Kappa, which is the best among the three polarization channels. Compared with VV and VH/VV channels, the VH polarization can better identify corn, wheat, and grass. Although the assessment metric shows that the VV polarization achieves best for the build-up region, many crops are misclassified as the build-up in Figure 6.

For statistical features, OGΓD $v$ and OGΓD $\sigma$ were tested. The OGΓD $v$ presented a low accuracy in all three polarization channels, especially for the three main crops in our study area. Nonetheless, another statistical feature, the OGΓD $\sigma$, performed best among the four tested features, showing the highest OA of 96.66% and 95.34% of the Kappa in the VH polarization channel. The OGΓD $\sigma^{VV}$ achieved a high precision for the corn and the soybean region, but only achieved 49.80% and 40.76% for wheat and grass, respectively. The OGΓD $\sigma^{VH/VV}$ was not as good at classifying crops as the VH and VV polarizations, while it was better at identifying build-up regions. Combing the visual and quantitative evaluation results, the OGΓD $\sigma^{VH}$ performed best among the four types of features.

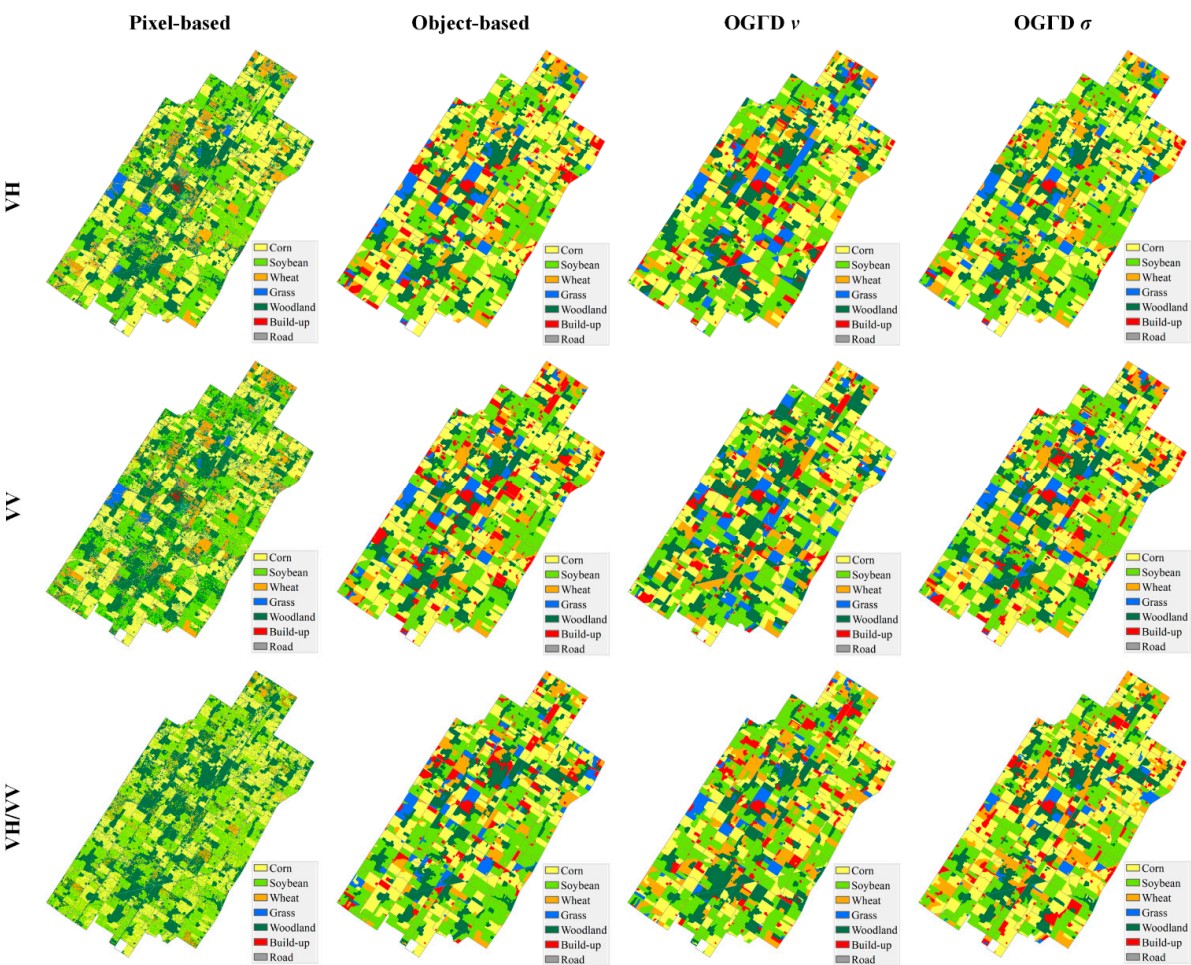

**Figure 6.** Crop classification results using a single type of feature. Each column is one feature type and each row is a polarization mode.

**Table 1.** The quantitative evaluation results of crop classification using a single type of feature.

| | Pixel VH | Pixel VV | Pixel VH/VV | Object VH | Object VV | Object VH/VV |
|---|---|---|---|---|---|---|
| **Corn** | 90.38% | 86.55% | 73.57% | 98.73% | 91.42% | 78.62% |
| **Soybean** | 88.55% | 80.40% | 71.06% | 87.97% | 90.50% | 100.00% |
| **Wheat** | 61.76% | 62.49% | 31.97% | 100.00% | 30.48% | 50.91% |
| **Grass** | 38.05% | 38.32% | 1.29% | 61.72% | 40.76% | 3.79% |
| **Woodland** | 93.56% | 84.89% | 91.15% | 96.93% | 100.00% | 94.33% |
| **Build-up** | 14.98% | 11.85% | 2.30% | 62.48% | 79.99% | 20.18% |
| **OA** | 85.21% | 79.29% | 69.42% | 92.17% | 87.39% | 83.79% |
| **Kappa** | 79.19% | 71.15% | 56.52% | 89.16% | 82.72% | 77.33% |
| | OGΓD $v^{VH}$ | OGΓD $v^{VV}$ | OGΓD $v^{VH/VV}$ | OGΓD $\sigma^{VH}$ | OGΓD $\sigma^{VV}$ | OGΓD $\sigma^{VH/VV}$ |
| **Corn** | 69.69% | 46.39% | 42.46% | 98.73% | 92.70% | 92.70% |
| **Soybean** | 62.27% | 43.27% | 67.72% | 94.29% | 95.38% | 90.17% |
| **Wheat** | 22.43% | 26.66% | 76.29% | 100.00% | 49.80% | 80.68% |
| **Grass** | 0.00% | 26.03% | 40.76% | 100.00% | 40.76% | 59.24% |
| **Woodland** | 39.30% | 57.46% | 72.51% | 100.00% | 90.03% | 74.58% |
| **Build-up** | 93.99% | 83.90% | 23.53% | 59.44% | 57.17% | 100.00% |
| **OA** | 56.59% | 46.22% | 58.54% | 96.66% | 88.25% | 86.56% |
| **Kappa** | 39.77% | 27.99% | 42.86% | 95.34% | 83.80% | 81.34% |

### 4.2. Crop Classification Using Feature Combinations

Compared with crop classification using a single type of feature, the feature combination can synthesize the advantages of different features, improving the accuracy and robustness of classification results. For pixel-based backscatter, object-based backscatter, and OGΓD $\sigma$, the VH polarization performed best among the three polarization channels, while VH/VV achieved best for OGΓD $v$. The first row in Figure 7 is the combinations of the backscattering features and statistical features in their best performance polarizations. Furthermore, the second row shows the combinations of all polarization channels for each type of feature. Table 2 is the quantitative assessment results.

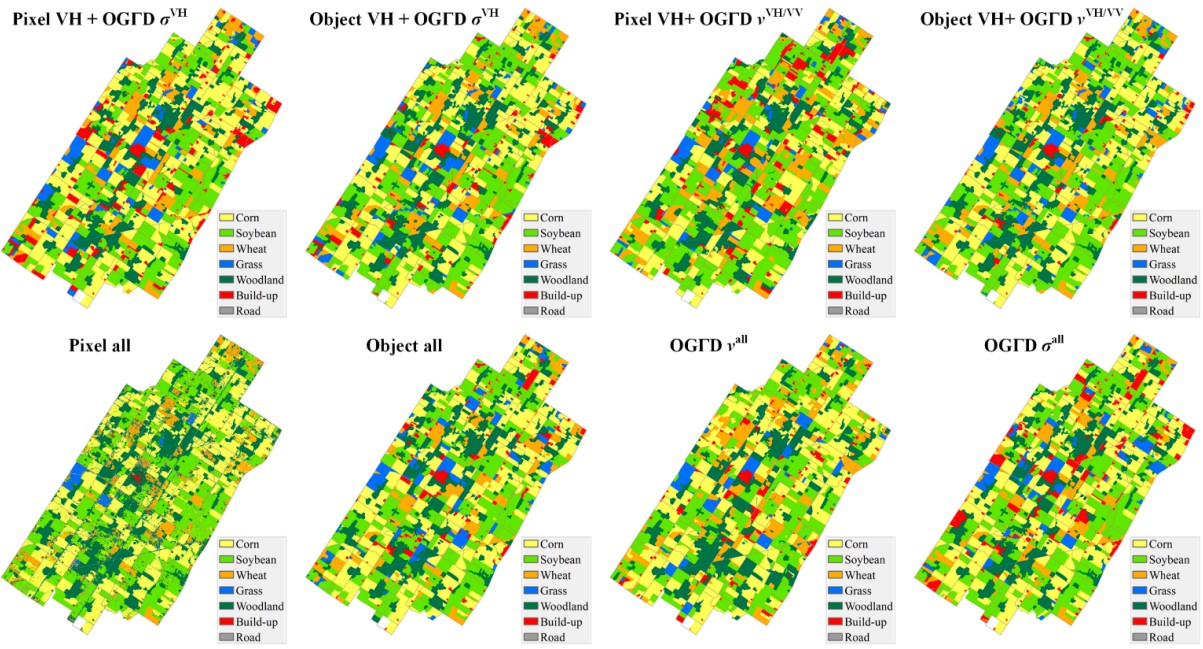

**Figure 7.** Crop classification results using feature combinations. The first row combines backscattering features and OGΓDs in the best-performance polarization channels. The second row combines different polarizations (VH, VV, and VH/VV) of the same type of features.

**Table 2.** The quantitative evaluation results of crop classification using feature combinations. The superscript of all means three polarization channels (VH, VV, VH/VV) were used.

|  | Pixel VH + OGΓD $\sigma^{VH}$ | Object VH + OGΓD $\sigma^{VH}$ | Pixel VH + OGΓD $v^{VH/VV}$ | Object VH + OGΓD $v^{VH/VV}$ | Pixel All | Object All | OGΓD $v^{all}$ | OGΓD $\sigma^{all}$ |
|---|---|---|---|---|---|---|---|---|
| **Corn** | 95.59% | 98.73% | 42.46% | 94.02% | 92.12% | 91.42% | 81.19% | 91.42% |
| **Soybean** | 87.97% | 94.29% | 67.72% | 87.97% | 90.18% | 100.00% | 81.20% | 94.29% |
| **Wheat** | 100.00% | 100.00% | 76.29% | 100.00% | 66.94% | 13.77% | 86.23% | 80.68% |
| **Grass** | 61.72% | 66.77% | 40.76% | 61.72% | 34.75% | 29.44% | 29.82% | 55.45% |
| **Woodland** | 96.93% | 100.00% | 79.70% | 100.00% | 93.98% | 96.93% | 80.79% | 100.00% |
| **Build-up** | 48.33% | 74.60% | 22.66% | 63.26% | 15.16% | 55.56% | 84.13% | 91.69% |
| **OA** | 90.84% | 95.54% | 59.85% | 91.18% | 86.58% | 88.58% | 79.22% | 92.03% |
| **Kappa** | 87.31% | 93.78% | 45.19% | 87.72% | 81.13% | 84.07% | 71.13% | 89.01% |

In Figure 7, the combinations of pixel-based backscatter and the statistical features were better for coherent speckle suppression compared with using pixel-based backscatter alone. However, the combination of pixel-based VH and OGΓD $v^{VH/VV}$ only achieved 45.19% of the Kappa, which was the lowest among all feature combinations. The two were not good when they were tested separately, and the stacking of errors led to poor classification results. By contrast, the combination of pixel-based VH and OGΓD $\sigma^{VH}$ was

much better, outputting 90.84% of the OA and 87.31% of the Kappa. A better result can be seen when combining object-based VH and OGΓD $\sigma^{VH}$, which are the two best single types of features, showing 95.54% of the OA and 93.78% of the Kappa. However, this accuracy is lower than using OGΓD $\sigma^{VH}$ alone, although more bands and information were added when using the feature combination.

The second row in Figure 7 combines different polarizations of the same type of feature, which means that it combines VH, VV, and VH/VV polarization channels for one type of feature. Although we synthesized all of the polarization channels, the pixel-based backscatter was affected by the speckle noise, obtaining 86.58% of the OA and 81.13% of the Kappa, which are higher than using the single-polarization band. Similarly, the classification results of OGΓD $\nu^{all}$ achieved 79.22% of the OA and 71.13% of the Kappa, which is better than any single polarization. In contrast, the other two types of features, object-based backscatter and OGΓD $\sigma^{all}$, show decreases when combining all polarizations. The OA for object-based backscatter dropped by 4% and the Kappa dropped by 5% compared to only using VH polarization, which can also be observed in OGΓD $\sigma^{all}$. For these two types of features, better crop classification results can be acquired using VH polarization alone.

### 4.3. Crop Sequence and Crop Rotation Mapping Result (2017–2021)

In the previous part, the classification performance of single types of features and feature combinations was tested, showing that the OGΓD $\sigma^{VH}$ achieved the best among them. In this section, the OGΓD $\sigma^{VH}$ was applied in five consecutive years (2017–2021) to obtain the annual crop maps, which are the basics for the crop sequence and crop rotation analysis. Figure 8 shows the crop classification results by combining the RF and the OGΓD $\sigma^{VH}$. The black rectangle region gives more details than the results of the whole area. Table 3 gives the assessment metrics, including PA and UA for each landcover type and OA and Kappa for each year.

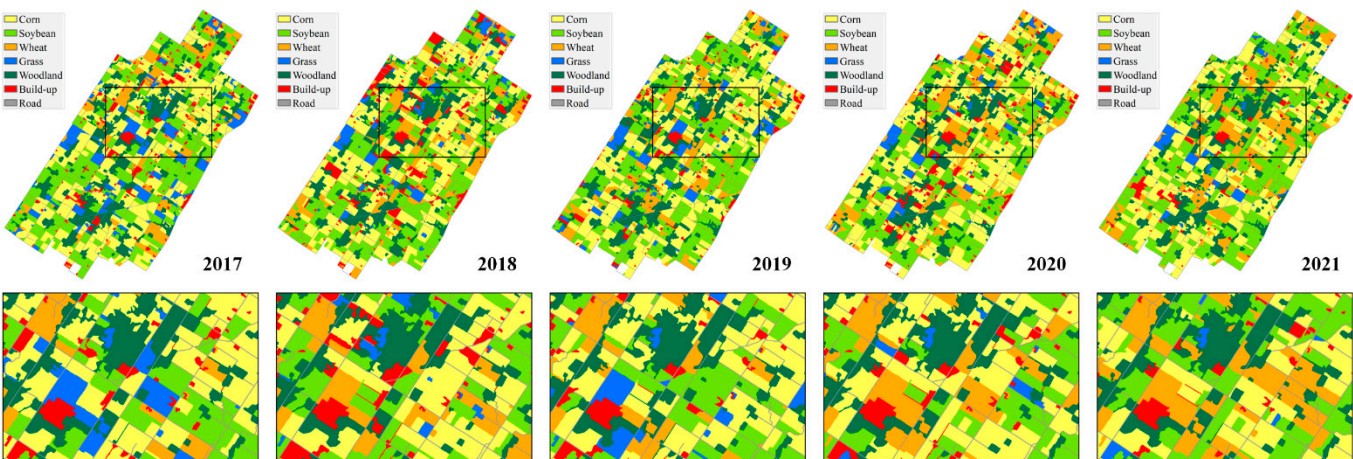

**Figure 8.** Annual crop mapping results from 2017 to 2021.

In Figure 8, overall, the classification results have good spatial-temporal consistency, especially for crops and the woodland regions. The overall accuracy is dominated by three main crop types: corn, soybean, and wheat. Thus, although several years show a lower precision for grass and build-up, the OA and the Kappa are also acceptable, e.g., 2018 and 2021. For 2021, the quantitative assessment indicates a misclassification between grass and build-up, which may be affected by the mixed pixel near the build-up regions. The best and worst years of the five crop maps are 2018 and 2019. For 2018, the UA for corn and soybean are 96.91% and 93.99%, and the OA and Kappa are 92.93% and 90.10%. The Kappa coefficients of all five years are higher than 90%, demonstrating that the results provide a reliable basis for subsequent crop rotation analysis.

**Table 3.** The quantitative evaluation results of annual crop maps from 2017 to 2021. Four metrics were selected, including producer's accuracy (PA), user's accuracy (UA), overall accuracy (OA), and Kappa.

|  | 2017 | | 2018 | | 2019 | | 2020 | | 2021 | |
|---|---|---|---|---|---|---|---|---|---|---|
|  | PA | UA | PA | UA | PA | UA | PA | UA | PA | UA |
| Corn | 97.27% | 95.90% | 96.91% | 100.00% | 98.73% | 99.72% | 97.20% | 93.41% | 100.00% | 100.00% |
| Soybean | 94.95% | 99.76% | 93.99% | 100.00% | 94.29% | 100.00% | 100.00% | 100.00% | 98.68% | 99.89% |
| Wheat | 82.27% | 87.70% | 100.00% | 58.26% | 100.00% | 78.45% | 44.44% | 57.54% | 100.00% | 76.58% |
| Grass | 75.91% | 51.81% | 32.28% | 63.45% | 100.00% | 100.00% | 77.72% | 100.00% | 50.63% | 100% |
| Woodland | 92.33% | 100.00% | 100.00% | 99.90% | 100.00% | 92.39% | 97.20% | 100.00% | 91.12% | 99.36% |
| Build-up | 94.64% | 53.03% | 98.20% | 27.02% | 59.44% | 73.38% | 100.00% | 59.82% | 56.96% | 22.29% |
| **OA** | 93.99% | | 92.93% | | 96.66% | | 95.14% | | 95.72% | |
| **Kappa** | 91.47% | | 90.10% | | 95.34% | | 92.80% | | 93.94% | |

After the integration of the five annual crop maps to a crop sequence layer, 561 individual areas with crop sequences are differentiated in the final dataset. The ten most frequent crop sequences from 2017 to 2021 are shown in Figure 9, in which S represents soybean, C represents corn, W represents wheat, and G represents grass. Table 4 lists the ten major crop sequences (in terms of total areas) that include a crop class in all of the five years.

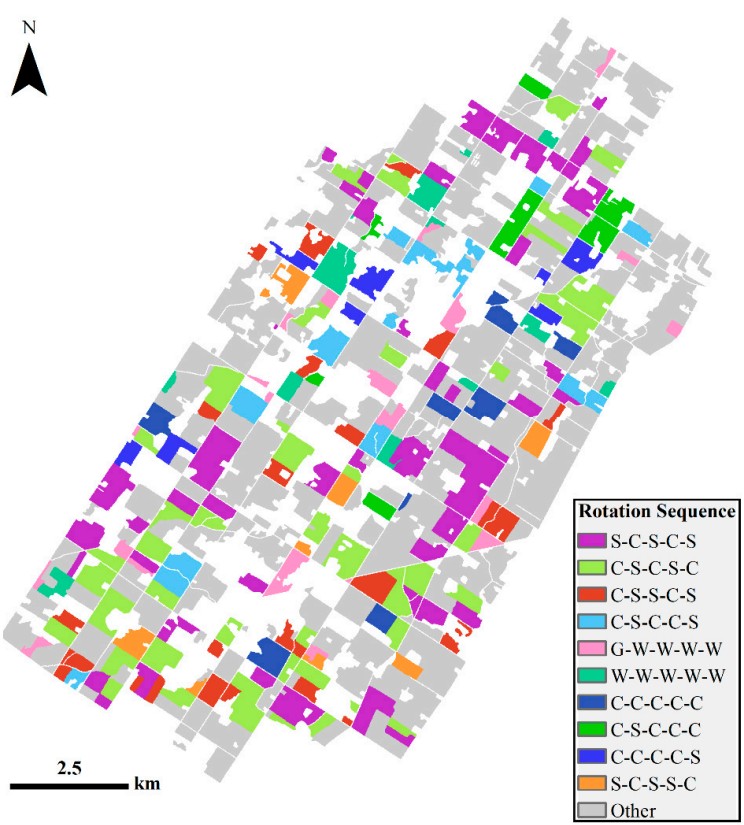

**Figure 9.** Map of selected crop sequences and crop rotations (2017–2021) out of the ten most frequent crop sequences. S = soybean, C = corn, W = wheat, and G = grass.

In Table 4, the first two crop rotation sequences (S-C-S-C-S and C-S-C-S-C) have the same crop rotation, i.e., soybean-corn and corn-soybean. These two sequences occupy 25.20% of the total area and are evenly distributed throughout the study area, which can be observed in Figure 10. The next two are C-S-S-C-S and C-S-C-C-S, which also consist of a similar crop rotation pattern. However, the order of planting was changed in some years, causing different crop sequences than the first two. For instance, the C-S-S-C-S includes the period of planting soybeans for two consecutive years, while the C-S-C-C-S includes two consecutive years of planting corn. The wheat planting in our study area is dominated

by a monoculture planting mode. The next two, G-W-W-W-W and W-W-W-W-W, consist of 2.55% and 2.51% of the total area, respectively. Another monoculture sequence is C-C-C-C-C, which has nine fields. Most of the fields in the study area are cultivated in a rotation pattern instead of a monoculture pattern, and corn and soybean are the two most important crops in this region.

**Table 4.** The ten major crop sequences of the study area (in terms of total field areas) from 2017 to 2021. S = soybean, C= corn, W = wheat, and G = grass.

| Rank | Crop Sequence | % of the Total Area | Cumulative % | Total Field Number |
|------|---------------|---------------------|--------------|---------------------|
| #1 | S-C-S-C-S | 12.80% | 12.80% | 50 |
| #2 | C-S-C-S-C | 12.40% | 25.20% | 50 |
| #3 | C-S-S-C-S | 4.34% | 29.54% | 29 |
| #4 | C-S-C-C-S | 3.66% | 33.20% | 14 |
| #5 | G-W-W-W-W | 2.55% | 35.75% | 21 |
| #6 | W-W-W-W-W | 2.51% | 38.26% | 14 |
| #7 | C-C-C-C-C | 2.50% | 40.76% | 9 |
| #8 | C-S-C-C-C | 2.08% | 42.85% | 8 |
| #9 | C-C-C-C-S | 1.97% | 44.81% | 8 |
| #10 | S-C-S-S-C | 1.93% | 46.75% | 9 |

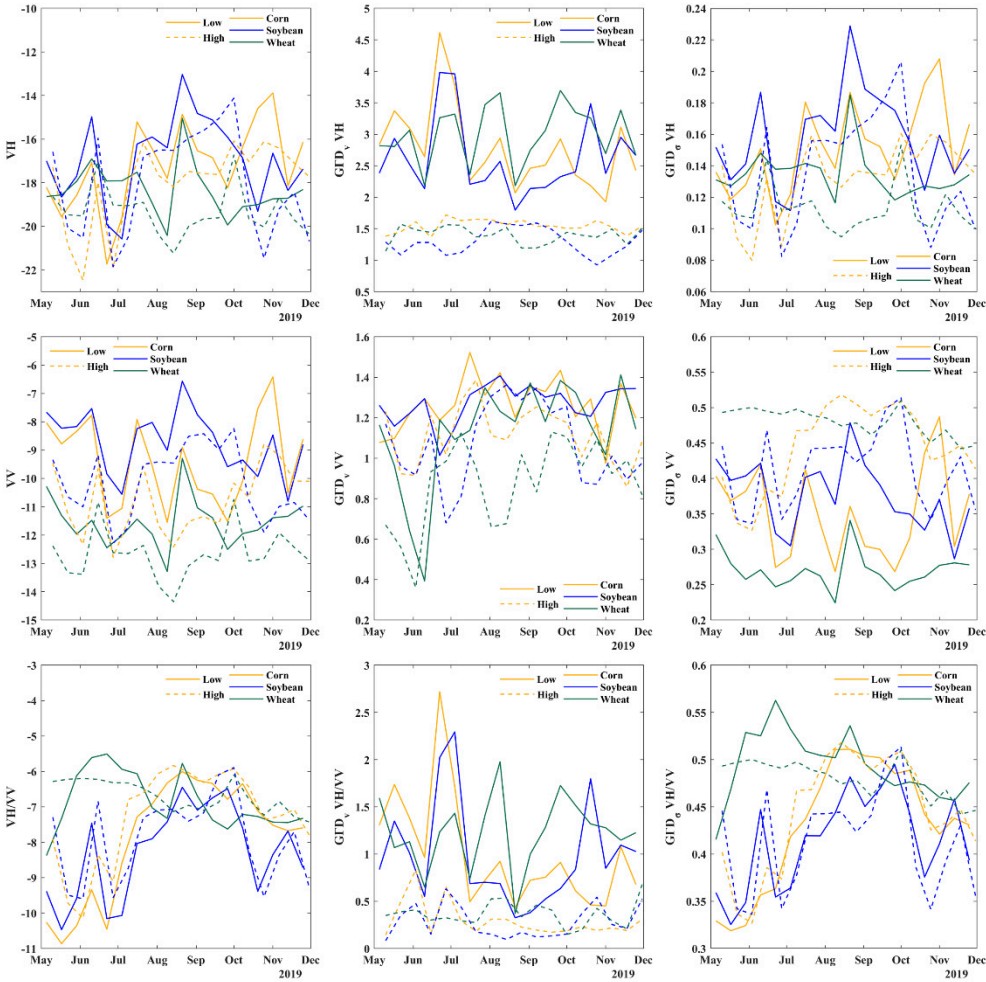

**Figure 10.** Feature time series for backscattering and OGI'Ds. The solid line represents a low incidence angle and the dashed line is a high incidence angle. The yellow, blue, and green represent corn, soybean, and wheat, respectively.

## 5. Discussion

### 5.1. Feature Time Series Analysis

In order to better understand the changes in the features of the crop-growing process, the features involving crop classification are analyzed in this part. The backscatter and OGΓDs in 2019 were selected as in the previous classification evaluation. The temporal variations of these features are shown in Figure 10. The solid line represents a low incidence angle (31.5°), and the dashed line is a high incidence angle (41.5°).

For all three polarization channels, the shapes of the backscattering time series and OGΓD $\sigma$ time series are in strong agreement showing similar trends, while OGΓD $v$ is totally different. For the soybean, the VH backscatter and OGΓD $\sigma$ increased before September. By contrast, the VV polarization was more stable during this season, showing a decreasing trend from early September to November, which was consistent with the VH polarization. During this period, as the soybean matured, the backscattering decreased with the vegetation water content. In addition, a valley can be observed in early July for both VH and VV channels. The SAR backscatter for the soybean consists of canopy scattering and soil surface scattering [31]. The canopy scattering of soybean, as well as of corn, must have increased due to the ongoing growth process during early July, while the total backscatter declined, which may have been caused by the reduction in the soil scattering due to drought. The valley was less significant than that of corn and soybean in the feature time series of wheat because the wheat was close to maturity and dominated by canopy scattering. In the time series of VH polarization, corn was showing an upward trend until November, which was consistent with its phenology, while the VV polarization fluctuated more obviously. This might be one of the reasons that the classification accuracy of VH polarization is higher than VV polarization in Table 1.

The influence of radar incidence angle was also discussed because there are two Sentinel-1 orbits in our study area. In Figure 10, for the backscatter time series, it can be seen that the backscatter of low-incidence angle images is higher than those of high-incidence angles, especially for the VV polarization. However, the difference between them is uneven, which is not only due to the radar being closer to the target at low incidence angles but also because the backscatter changes with the growing state of the crop. It is worth noting that September to November is a good time window for crop classification due to the maximum difference of backscatter between the three crop types. By contrast, OGΓDs are more sensitive to radar incidence angles, especially for the $v$ in the VH polarization and $\sigma$ in VV polarization. Therefore, the low incidence angle has a higher value of OGΓD $v$, while the value of OGΓD $\sigma$ of the low incidence angle is lower than the high incidence angle.

### 5.2. Feature Importance Evaluation

Due to the time-series features of SAR data used in this research, the feature importance evaluation aims to discover the contribution of different features in different crop growth stages. Figure 11 gives the variations in feature importance with the date and output by the RF model. Both low and high incidence angle data are plotted by the solid line and the dashed line, respectively. The first aim of evaluating the features' contribution is to better understand how feature time-series work for crop classification. The second aim is to provide a reference for future research on crop classification that requires feature selection when using multi-temporal or time-series Sentinel-1 images.

In Figure 11, for the pixel-based VH backscatter, the highest feature importance appears from October to November corresponding to the corn harvest in this period. The harvesting behavior at this stage makes it different from other landcover types in SAR images, and since corn is one of the main crop types in the study area, the feature importance from October to November presents high values. The peak can also be seen in the results of Object-based VH and OGΓD $\sigma^{VH}$, which are slightly lower than the Pixel-based one. The period with the greatest contribution to the high incidence angle results is between September and October. At this stage, the soybean gradually matured and the backscattering value continued to decrease, as shown in Figure 10, which helped the

classifier to identify it with other crop types. It can be seen that there are differences in the sensitivity of different incident angles to crops in VH polarization. Low incidence angles are more sensitive to corn, while high incidence angles are more sensitive to soybean. Although similar peaks can be observed for the results of the other two polarization modes, the period of those peaks of feature importance differs from VH polarization. Despite the results of OGΓD $v$ showing that it is helpful for crop classification during the whole period, the feature importance values remained low over the whole crop growing season. The OGΓD $v$ may indicate some temporal invariant characteristics of the crops, whereby the OGΓD $v$ exhibits uniform importance throughout the crop growing season. The feature importance assessment was consistent with the feature time-series analysis in the previous section, especially for the VH polarization, which was mainly affected by the different phenological stages of the crops.

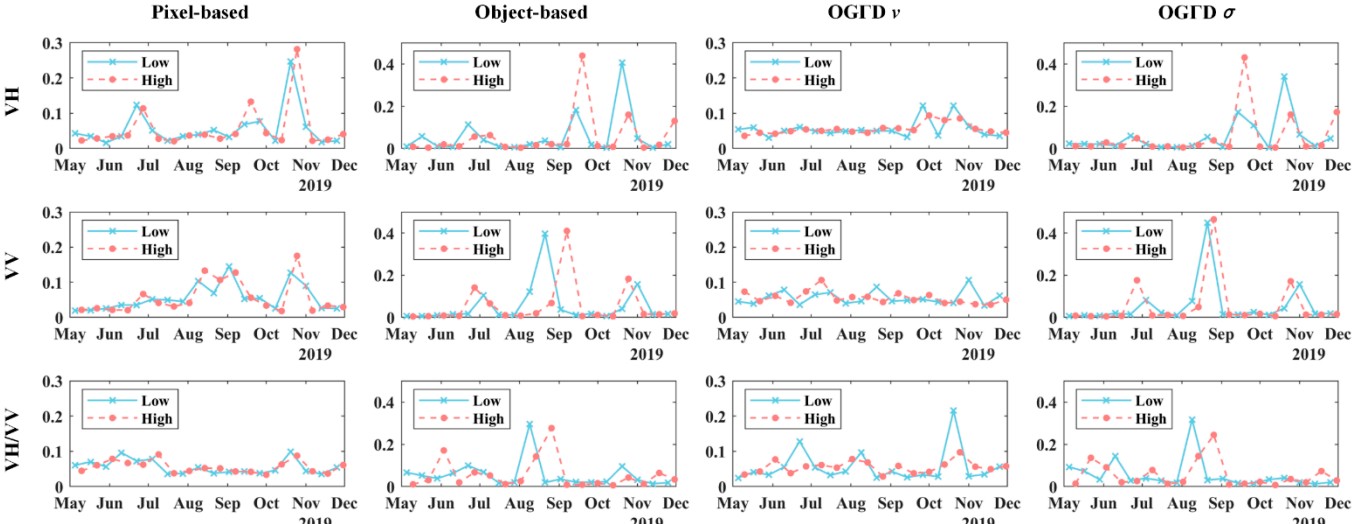

**Figure 11.** The results of feature evaluation. The solid line represents a low incidence angle, and the dashed line is a high incidence angle. Each row represents a polarization (VH, VV, VH/VV) and each column represents a feature type. The *x*-axis is the date and the *y*-axis is the feature importance value.

*5.3. Classification Performance Using the Automatic Segmentation*

When applying the method to a larger scale, an automatic segmentation scheme that can achieve high accuracy of segmentation results is required. This paper aims to explore the potential of statistical features instead of developing a new segmentation approach. Here, we will evaluate the performance of the automatic segmentation method on crop classification results. The segmentation algorithm used is an adaptive-scale approach that can avoid the influence of parameter selection on the results [32]. Figure 12 shows the annual crop classification results using automatic segmentation, and Table 5 shows the quantitative evaluation results.

Compared with the classification results using the manual segmentation method (Figure 9 and Table 3), the crop maps based on automatic segmentation have rougher boundaries caused by the segmentation method. Due to the impact of inherent coherence speckles in SAR images, automatic segmentation results cannot precisely match the edges of fields leading to errors in the subsequent classification steps. In Figure 12, some small misclassification can be seen in the interior of the fields. The automatic segmentation algorithms divided the small heterogeneous areas into two parts with the farmland. In the subsequent classification, these small blocks cannot obtain the global information, resulting in the wrong types. The quantitative results also demonstrated that the automatic method has a lower overall accuracy than using the manual delineated results, especially for the build-up regions. Small artificial structures on the edge of farmland are difficult to identify for the automatic method. The accuracy might be improved if a better automatic

segmentation method can be introduced. However, instead of directly transferring a general segmentation method, it requires the designing of a new segmentation approach that considers the characteristics of the agricultural area, such as the boundary rules of the fields and the heterogeneity of the SAR images.

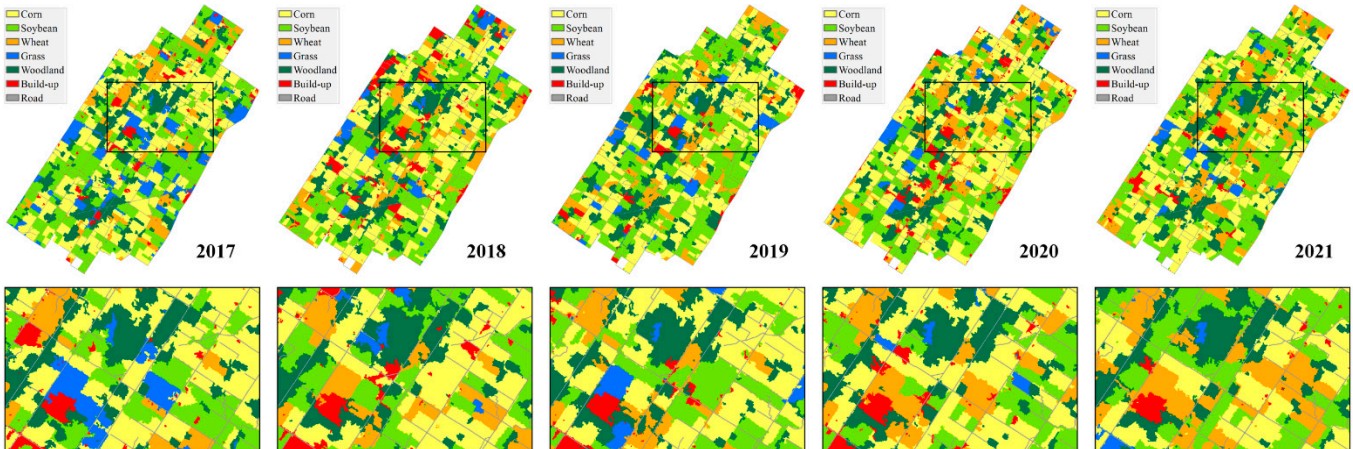

**Figure 12.** Annual crop mapping results from 2017 to 2021 based on automatic segmentation.

**Table 5.** The quantitative evaluation results of annual crop maps from 2017 to 2021 based on automatic segmentation. Four metrics were selected, including producer's accuracy (PA), user's accuracy (UA), overall accuracy (OA), and Kappa.

|  | 2017 | | 2018 | | 2019 | | 2020 | | 2021 | |
|---|---|---|---|---|---|---|---|---|---|---|
|  | **PA** | **UA** | **PA** | **UA** | **PA** | **UA** | **PA** | **UA** | **PA** | **UA** |
| Corn | 89.67% | 92.44% | 93.31% | 93.24% | 94.59% | 93.13% | 93.12% | 89.26% | 95.56% | 93.76% |
| Soybean | 94.24% | 93.06% | 87.62% | 96.27% | 89.33% | 93.58% | 93.53% | 95.36% | 93.19% | 92.89% |
| Wheat | 79.01% | 79.68% | 91.58% | 52.11% | 74.10% | 54.70% | 42.23% | 38.97% | 94.40% | 66.06% |
| Grass | 65.31% | 39.64% | 30.51% | 57.16% | 84.25% | 76.93% | 43.41% | 74.24% | 48.62% | 87.23% |
| Woodland | 76.07% | 87.53% | 86.13% | 89.17% | 83.92% | 85.36% | 82.01% | 90.05% | 76.14% | 91.91% |
| Build-up | 34.98% | 22.97% | 32.21% | 9.02% | 35.98% | 52.48% | 44.82% | 19.32% | 10.35% | 5.13% |
| **OA** | 86.81% | | 86.14% | | 88.10% | | 87.53% | | 89.07% | |
| **Kappa** | 81.32% | | 80.59% | | 83.38% | | 81.58% | | 84.46% | |

## 6. Conclusions

This paper explored the potential of Sentinel-1 GRD data for crop classification and crop rotation mapping using proposed statistical features named OGΓDs and the RF algorithm. First, the SAR time-series stacks were established, including VH, VV, and VH/VV polarization channels. Both pixel-based features at the pixel scale and object-based features at the field scale were generated, including pixel-based backscatter, object-based backscatter, OGΓD $v$, and OGΓD $\sigma$. These features were input into the RF model to evaluate their performance. Then, the OGΓD $\sigma^{VH}$ that achieved the best performance when compared with other features and feature combinations was selected to produce the annual crop maps from 2017 to 2021. Finally, the annual crop maps were synthesized as a crop sequence dataset, and the crop rotations were given. Some key aspects of the experiment result are (i) the OGΓD $\sigma^{VH}$ achieved 96.66% of the OA and 95.34% of the Kappa, improving by around 4% and 6%, respectively, compared with the object-based backscatter in VH polarization; (ii) the crop sequence soybean-corn-soybean-corn-soybean (together with corn-soybean-corn-soybean-corn) has the highest count with 100 occurrences (25.20% of the total area); (iii) the top ten crop sequences are dominated by corn and soybean, and the soybean-corn (together with the corn-soybean) is the most representative crop rotation in our study area.

The feature time-series analysis and the importance evaluation found that the time-series SAR data had correlations with the phenology of crops, which may be helpful to feature selection when promoting the method to a larger scale. Images of crop maturity and harvest are more important than images of other periods and are the preferred option when data for the entire season are not available. Moreover, the proposed method is compatible with automatic segmentation, which needs to be improved to suppress the errors in the segmentation step. The new proposed high-accuracy classification approach based on the SAR time-series backscatter and statistical features offer some new insights into crop rotation monitoring, giving the basic data for government food planning decision-making.

**Author Contributions:** Experiment and initial draft preparation, X.Z.; review and editing, J.W.; editing and formal analysis of the paper, Y.H. and B.S. All authors have read and agreed to the published version of the manuscript.

**Funding:** This research was funded by the Natural Science and Engineering Research Council of Canada (NSERC) Discovery Grant (grant number RGPIN-2022-05051) awarded to Jinfei Wang.

**Acknowledgments:** The authors would like to thank the EU Copernicus Program for providing the Sentinel-1 SAR data. We would also like to thank Chunhua Liao, Qinghua Xie, and Minfeng Xing for their work on ground truth data collection. Thanks to Deema Hashim for her help with proofreading. Thanks to Junxiong Zhou and Huashi Zheng for their support. Special thanks to the editors and reviewers for their comments to improve the article.

**Conflicts of Interest:** The authors declare no conflict of interest.

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
