# Peer review of "Crop Classification and Representative Crop Rotation Identifying Using Statistical Features of Time-Series Sentinel-1 GRD Data"

_remotesensing, doi:10.3390/rs14205116_

Round 1

Reviewer 1 Report

The paper proposed a method for mapping crop types and rotation patterns in a small area, using SAR data and statistical features. It provided a clear background about previous studies and the importance of conducting the proposed study. The paper is relatively well presented in describing the data and methods used and presenting the results.

However, I have two major concerns which I think the paper requires some extra analysis before it can be published.

The first is the paper lacks in-depth discussion about proposed methods and features. The current 5.1. and 5.2 sections read more like Results to me. While these figures and findings should move to the Result section, it would make more sense if the authors could provide their insights about these figures / findings. For instance, in Figure 11, why are the peaks (the features at those days) stands out with higher importance value? This would also responds to the authors’ statement of using RF only in this study (L249-252), which is aimed to evaluate the indices contributions?

My second concern is the author proposed to manually segment the objects. Did the authors delineated the objects annually? Seems a ‘yes’ to me from Figure 9, where large field in one year was identified as several small fields in another? Manually delineate the objects seems fine for a small area, however, I suppose one of the main reasons of using RS for crop classification is to out-scale the method to larger areas. Therefore, it would be better if the authors could add analysis of using automated segmentation method and then apply the proposed methods?

Also, I suppose the segmentation would also affect the rotation results. what if one field (as a whole) was planted as C in one year, and then in the following years the field been planted with more than one crops (e.g. C, S and W)?

Reviewer 2 Report

see the attachment

Reviewer 3 Report

This is an interesting study in using SAR for crop classification. The manuscript is overall well written. But I sill have some questions related to the methods that need to be clarified. 

Line 42-45: Lack of references.

Line 70: "The experimental results showed that the SAR data could improve the classification accuracy compared with using optical satellites alone." Lack of references.

Line 149: What is "crop inventory"?

Figure 2. Why is the validation set have much fewer grass and woodland pixels? It seems unbalanced. How did you separate the data for training and testing? 

Line 274. Why do you use 2019-05-03 as an example for the features? The crops are just barely planted in early May.

Table 3: What are "PA" and "UA"?

Figure 11: Did you train the model separately for each year? If so, how do we know if the model can be used in predicting the coming year before the ground truth are available? 

Round 2

Reviewer 2 Report

Well improved. No further comments from my  end

Author Response

Thank you very much for your efforts and patience in improving our article. We thoroughly checked our spell according to the requirements. Please check the manuscript.